# OpenReview forum: "Iterative Training of Language Models with Opponent Modeling for Red Teaming Data Generation"
_ICLR.cc/2025/Conference — Submitted to ICLR 2025_

### Official Review · Reviewer_JRUT · 2024-10-27

**Soundness:** 2
**Presentation:** 2
**Contribution:** 2
**Rating:** 5
**Confidence:** 4

**Summary:**

This paper introduces a method to iteratively attack (jailbreak) the LLMs, aka red-teaming. It first validates that iteratively refining the prompt for jailbreaking improves ASR (attack success rate) using large models like Llama3-70B-chat. Considering that using large model is costly and it only relies on the models' zero-shot ability, the paper further suggests a model training method which turns small model like Llama3.1 8B to generate jailbreak prompts that can be used for red-teaming.

Key idea in the model training is called opponent modeling -- the paper introduces three loss objectives that lead to (1) opponent loss: generate responses as an opponent, (2) topic-aware attack loss: conditioned by topic and generate attacks, and (3) refine attack loss: refine the prompt based on initial prompt, topic, and opponent response. Iterative training with these objectives lead to higher ASR with the trained model, but it sacrificied the diversity prompts.

The paper further showed that simple pruning doesn't hurt the ability to generate red teaming prompts for trained models -- which the authors compare with the prior work (Wei et al, 2024) that safety alignment and red teaming abilities have distinct properties in terms of model parameter spaces.

**Strengths:**

The motivation of the work is clear that existing jailbreaking works that use LLMs can be cost-inefficient -- existing works rely a lot on large models to generate attacks. I like the idea to allow small models to also generate red teaming prompts.

I think the main idea of the work is iterative training, and the objectives introduced here are interesting and novel. The experimental results (in Table 3) clearly showed that (1) more iterations of training lead to higher ASR, (2) it can even surpass ASR of the dataset that is used to train the generator, and (3) refinement further improves ASR.

**Weaknesses:**

- I think the main weakness of the paper is lack comparison with state-of-the-art jailbreaking strategies, and this may lead to weaker impact to the field. Specifically, there are some representative attacks like GCG, TAP, PAIR, and AutoDAN that can set a strong baseline for attacks. Comparing ASR with these numbers will be necessary to show the effectiveness of the proposed approach for training models to generate attacks, even though they require using larger models. Presenting more baseline numbers will help a lot to understand the results.

- Though the training objectives are novel, the process for model training is complex and the main reason for this is that it includes multiple stages during the training. I believe streamlining the training stages could be more practical and helpful for people who want to apply this: for example, I don't understand the effect of each loss objectives. Detailed ablations might help understanding the meaning of each loss objectives; especially, when I see Table 2, using response as a context does not significantly improves ASR. This made me think skeptical to use these complex loss objectives for model training.

Additional comments:

- L286: reference for `See in Section` is missing.
- Personally, I couldn't understand why Section 3.4 is included in the main paper -- I think it is not highly relevant to the main contribution of this work. I would recommend moving this content to appendix and rather introduce more experimental results such as ablations and comparing with existing jailbreak baselines.

**Questions:**

My questions are aligned with the contents provided in Weaknesses section. In particular:

- Could you compare with existing jailbreaking baselines to show how the ASR obtained from your trained models are effective enough?
- Could you provide ablation studies results to confirm that the different stages in the iterative training processes are all necessary?

---

### Official Review · Reviewer_nGvu · 2024-10-27

**Soundness:** 3
**Presentation:** 3
**Contribution:** 3
**Rating:** 6
**Confidence:** 4

**Summary:**

This paper introduces an iterative training framework for generating red teaming data to test and evaluate Large Language Models' (LLMs) safety mechanisms. The approach combines three key strategies - direct prompting, mutation strategies, and persuasion techniques - along with opponent modeling to generate adversarial prompts. The method demonstrates significant success in bypassing safety measures, achieving attack success rates of 74.95% on Llama2-7b-Chat and 69.10% on Llama3-8b-Instruct. The research also reveals that red teaming capabilities are densely distributed across model parameters, unlike safety alignment features which tend to be more sparsely represented.

**Strengths:**

1， The iterative training framework shows particular effectiveness in distilling red teaming capabilities from larger models to smaller ones, making it more practical for deployment.
2， The empirical analysis provides valuable insights into the fundamental nature of adversarial capabilities in language models and how they differ from safety alignment features.

**Weaknesses:**

1, The methodology shows limitations in handling diversity collapse during the training process. Although opponent modeling helps mitigate this issue, the problem persists to some degree, suggesting the need for more robust solutions to maintain prompt diversity throughout the training process.
2, the computational requirements of the approach, particularly in the data generation phase, could limit its practical applicability in resource-constrained settings. The need for multiple iterations and large-scale model training might make it challenging to implement in some contexts.
3, Some questions leave unanswered about the long-term effectiveness of the generated adversarial prompts and whether language models might develop resistance to these attack strategies over time through further safety alignment training.

**Questions:**

NA

---

### Official Review · Reviewer_zzR3 · 2024-11-04

**Soundness:** 2
**Presentation:** 2
**Contribution:** 2
**Rating:** 3
**Confidence:** 3

**Summary:**

The paper proposes a framework to generate a red-teaming dataset. The first part of the framework involves investigating the performance of directly using LLMs for red-teaming prompt generation. It employs three strategies: 1) refining adversarial prompts by directly prompting an LLM (Llama3-70b-Chat); 2) mutating prompts using mutation rules (such as Sentence Rearrangement and Style Transfer); and 3) making prompts more persuasive using persuasion techniques (selected from 40 techniques outlined in the PAP work). The approach also considers enhancing the red-teaming capability of an attacker LLM through iterative training, with or without knowledge of the opponent's responses. Modeling the responses of attackers does contribute to generating more effective jailbreak prompts. The introduction of opponent modeling loss to train the attacker model is a novel contribution.

**Strengths:**

**Strengths**

The main strength of the paper is a new way to train an attack module. In my understanding, existing works don't consider modeling the behavior of the opponent (target). Results also bolster the idea of modeling the opponent which can spark future red-teaming research.

**Weaknesses:**

**Weaknesses:**
- The introduction is somewhat weak in terms of motivation. The three strategies proposed exist in various forms in the jailbreak literature, so it becomes even more important to properly highlight the contributions and their significance.
- There is a significant amount of related literature and compelling baselines that could have been used to compare the effectiveness of the approach. One such area of research is quality-diversity search, such as rainbow teaming [R1] and its successors [R2, R3]. These works train attacker modules to generate more harmful prompts and use prompt-response safety classifiers to iteratively obtain a more harmful set of diverse prompts. Additionally, there are more relevant baselines [R5] for ASR comparison.
- That said, I believe rainbow-teaming [R1, R2, R3] and wild-teaming [R4] (see Table 2) provide a better ASR both with and without knowledge of the opponent. I am not sure how the proposed approach is advancing the field of red-teaming in any aspect, whether it be proposing a better red-teaming dataset or target-aware red-teaming.
- Moreover, the paper lacks a more detailed analysis across different families of models. It would be interesting to see the ASR results on Llama, Mistral, GPT, and Claude family models.

Essentially, the paper lacks proper positioning, comparisons against compelling baselines, and an extensive analysis across different families of models. I am leaning towards rejection as of now.

References:

[R1] Samvelyan, Mikayel, et al. "Rainbow teaming: Open-ended generation of diverse adversarial prompts." arXiv preprint arXiv:2402.16822 (2024).

[R2] Han, Vernon Toh Yan, Rishabh Bhardwaj, and Soujanya Poria. "Ruby Teaming: Improving Quality Diversity Search with Memory for Automated Red Teaming." arXiv preprint arXiv:2406.11654 (2024).

[R3] Deep Pala, Tej, et al. "Ferret: Faster and Effective Automated Red Teaming with Reward-Based Scoring Technique." arXiv e-prints (2024): arXiv-2408.

[R4] Jiang, Liwei, et al. "WildTeaming at Scale: From In-the-Wild Jailbreaks to (Adversarially) Safer Language Models." arXiv preprint arXiv:2406.18510 (2024).

[R5] https://github.com/EasyJailbreak/EasyJailbreak

**Questions:**

Edits:

- Line 107, please define blue-team (and red-team) LLM in the context of this work.

- Line 286 "See in Section" looks incomplete.

- Equation (1) can be expressed in a better form, or include a 1-2 line explanation on it. The same goes for Algorithm 1, the draft text is only elaborated until line 235, later part of the algorithm isn't covered in the main text.

---

### Official Review · Reviewer_Q6JU · 2024-11-05

**Soundness:** 2
**Presentation:** 2
**Contribution:** 2
**Rating:** 3
**Confidence:** 3

**Summary:**

The paper presents a red-teaming framework designed to improve the robustness of large language models (LLMs) through iterative adversarial prompt refinement. It introduces a smaller model, Llama3.1-8B, with opponent modeling capabilities to simulate responses and optimize attack effectiveness, addressing challenges of limited diversity and computational costs in traditional red-teaming. The framework achieves high attack success rates on Llama models while maintaining prompt diversity, suggesting that red-teaming abilities are densely embedded in model parameters, distinct from the sparsity in safety alignment. The authors provide code and data to support future LLM safety research.

**Strengths:**

1. Efficient Adversarial Prompt Generation: The framework enables iterative prompt refinement with opponent modeling, improving the attack success rate and reducing dependence on human involvement in adversarial prompt crafting.
﻿
2. Enhanced Prompt Diversity: By integrating opponent modeling, the model preserves prompt diversity, addressing the common issue of repetitive adversarial prompts.
﻿
3. Parameter-Level Insights: The study provides valuable insights into how red-teaming abilities are distributed within LLM parameters, enhancing understanding of safety-critical neurons and their distinct behavior compared to safety alignment features.

**Weaknesses:**

1. Insufficient Background on Red Teaming: The paper lacks a detailed introduction to the concept and significance of red teaming, limiting comprehension for readers unfamiliar with this approach.
2. Conceptual Ambiguity: Some core concepts and methods are not well-defined, leading to confusion about certain techniques and their applications within the red-teaming process.
3. Incomplete Methodology Explanation: The framework's methodology lacks necessary conceptual and mathematical clarity, with insufficient formulas and step-by-step explanations, which makes the approach challenging to reproduce or understand in depth.

**Questions:**

None

---

### Meta-Review · Area_Chair_9yYt · 2024-12-20

**Metareview:**

This paper presents an iterative training framework for generating red teaming data to evaluate the safety mechanisms of Large Language Models (LLMs). The approach combines direct prompting, mutation strategies, and persuasion techniques with opponent modeling to create adversarial prompts. The experimental results show good performance of the approach.

Although the paper is fairly well written and easy to follow, there are several major concerns regarding insufficient background on Red Teaming, conceptual ambiguity, incomplete methodology explanation and lack of detailed analysis. Given that there is no response from the authors, I am recommending rejection of the work.

**Additional Comments On Reviewer Discussion:**

Although the paper is fairly well written and easy to follow, there are several major concerns regarding insufficient background on Red Teaming, conceptual ambiguity, incomplete methodology explanation and lack of detailed analysis. Given that there is no response from the authors, I am recommending rejection of the work.

---

### Decision · Program_Chairs · 2025-01-22

Reject